# Decreasing Hypothermia-Related Escalation of Care in Newborn Infants Using the BEMPU TempWatch: A Randomised Controlled Trial

**DOI:** 10.3390/children8111068

**Published:** 2021-11-19

**Authors:** Donna Lei, Kenneth Tan, Atul Malhotra

**Affiliations:** 1Department of Paediatrics, Monash University, Melbourne, VIC 3168, Australia; dlei0002@student.monash.edu (D.L.); kenneth.tan@monashhealth.org (K.T.); 2Monash Newborn, Monash Children’s Hospital, Melbourne, VIC 3168, Australia; 3The Ritchie Centre, Hudson Institute of Medical Research, Melbourne, VIC 3168, Australia

**Keywords:** breastfeeding, low birth weight, parents, small for gestational age, temperature control

## Abstract

Objective: To determine whether incorporating BEMPU TempWatch into the care of LBW/SGA neonates for continuous temperature monitoring decreases the rate of hypothermia requiring escalation of care. Methods: This was a randomised controlled trial conducted in a tertiary hospital in Melbourne, Australia. Participants were late preterm and term LBW/SGA neonates on the postnatal wards. Neonates were randomly assigned to receive either the BEMPU TempWatch in addition to standard care, or to receive standard care alone for the first 28 days of life. The primary outcome was hypothermia requiring escalation of care during initial hospital stay after birth. Results: Trial was discontinued after planned interim feasibility analysis, due to very low rates of hypothermia requiring escalation of care. In total, 75 neonates were included, with 36 in the intervention (TempWatch) group and 39 in the control group. The rate of hypothermia requiring escalation of care was 2/36 (5.6%) in the TempWatch group and 1/39 (2.6%) in the control group (relative risk (RR) 2.17, 95% CI 0.21 to 22.89). Rates of exclusive breastfeeding at discharge were 22/36 (61.1%) in the TempWatch and 13/39 (33.3%) in the control group (RR 1.83, 95% CI 1.10 to 3.07, *p* = 0.02). All other secondary outcomes were similar between the groups. Conclusions: Low rates of hypothermia requiring escalation of care in a tertiary, high-income setting meant it was not feasible for studying the effects of the TempWatch for this outcome. TempWatch may have a role in promoting exclusive breastfeeding, and this needs to be explored further.

## 1. Introduction

Low birth weight (LBW) and small for gestational age (SGA) babies are at a greater risk of hypothermia due to numerous factors that either increase their heat loss [1] or reduce their ability to produce heat to maintain normothermia [2] compared with normal birth weight and appropriate for gestational age neonates. Neonatal hypothermia is associated with significant morbidities and mortality. Early detection of hypothermia allows for prompt intervention with conservative warming measures and may reduce the need for invasive warming measures, including placement on a heated mattress, under a radiant warmer, or in an incubator. These measures are costly [3], pose risks to the baby including overheating [4] and dehydration [5], and separate babies from their mothers [6]. Currently, many methods of temperature monitoring in neonates are available [7]. Many of these have variable accuracy, suboptimal efficacy, and are unable to continuously monitor the neonate.

BEMPU TempWatch (BEMPU Health, Bangalore, Karnataka, India) is a novel single-use device that allows for continuous temperature monitoring of neonates for up to 30 days [8]. It consists of a thermistor metal cup within a plastic casing and a silicone band that can be worn around the wrist of neonates (Figure 1) who weigh 800–3300 g (BEMPU technical specifications). The TempWatch is powered by a battery to avoid the need for an external power supply [9]. When it detects that the neonate’s temperature is less than 36.5 °C, the TempWatch provides an audio–visual alarm to encourage carers to provide skin-to-skin contact (SSC) to their babies [8]. It is targeted for use in low- and middle-income countries (LMICs), with its simple-to-use design intended to be understood even by parents with low education levels who may be unaware of their baby’s risk of hypothermia [9].

Studies evaluating the BEMPU TempWatch in LMICs have shown that it is highly accurate at detecting hypothermia [10], promotes neonatal weight gain and SSC, and causes no adverse effects in the neonate [11,12]. However, its usefulness in a high-income country setting has not been evaluated. Given the differing levels of education and resources available between high-income countries and LMICs [13], it was important to determine whether the benefits of the TempWatch shown in LMICs also applied to a high-income setting. This study examined the effects of incorporating the BEMPU TempWatch into the routine care of LBW/SGA neonates to determine whether continuous temperature monitoring of these babies reduced the rate of hypothermia requiring escalation of care. We hypothesised that using the TempWatch would allow for earlier detection of hypothermia and intervention with conservative warming measures, therefore reducing the need for the escalation of care to invasive warming measures.

## 2. Methods

### 2.1. Study Design and Setting

This was a single-centre prospective randomised controlled trial conducted in a tertiary hospital (Monash Medical Centre) in Melbourne, Australia.

### 2.2. Patients

Neonates were eligible for inclusion if they were LBW (birth weight < 2500 g) and/or SGA (birth weight < 10th centile), born at 35 weeks’ gestation or greater, and were being cared for in the postnatal wards of the hospital. Neonates who were admitted to the special care nursery (SCN) or neonatal intensive care unit (NICU) soon after birth were excluded, unless they were returned to the postnatal wards within 48 h of life. Reasons for neonates requiring SCN admission soon after birth included prematurity of less than 35 weeks’ gestation and birth weight less than 2000 g. Neonates were also admitted to the SCN if they were at a high risk of sepsis, or experienced persistent hypothermia or prolonged hypoglycaemia refractory to increased feeds and Glucogel. Neonates were also admitted to the SCN and/or NICU if they had other morbidities, which were unable to be managed on the postnatal wards, such as respiratory distress syndrome, congenital malformations, and cardiac abnormalities.

The main reason for eligible families not being approached was if the research team was unavailable during the time the neonate was eligible for recruitment. Furthermore, families were not approached if neither parent had a level of English that allowed them to understand the verbal explanation of the project and the written information provided in the parent information and consent form. Eligible families were also not approached if there were significant concerns regarding the care of the baby, including maternal substance abuse, poorly managed maternal mental health conditions, or child protection issues.

### 2.3. Recruitment and Consent

Recruitment for the study occurred between March and August 2021. The families of neonates who fulfilled the inclusion criteria were invited to participate in this study in the postnatal wards by the principal investigators (D.L. and A.M.). Eligible families were approached any time from when mother and baby were transferred to the postnatal wards from the delivery suite, to within 48 h after delivery. Written informed consent was then obtained from one or, where possible, both parents.

### 2.4. Randomisation

Neonates were randomly allocated (1:1 group allocation) to either the intervention (TempWatch) or control group in block randomisation groups of 12. The randomisation order was computer-generated, and then placed in sequentially numbered opaque sealed envelopes by an independent research nurse. Once parents consented to their baby’s participation in the study, the principal investigator opened the randomisation envelope corresponding to the neonate’s study number to reveal their allocated group and randomised the neonate accordingly.

Neither the participating family nor the principal investigator was blinded to the neonate’s group allocation, as no placebo watch was available.

### 2.5. Intervention

Infants were randomly allocated to either the TempWatch or control group. The TempWatch group received the BEMPU TempWatch in addition to standard neonatal care. At randomisation, the parents of neonates who were allocated to this group were provided with the TempWatch. They were instructed to use the device for their baby’s first month of life, both in hospital and at home, and to provide SSC to their babies whenever hypothermia was indicated by the TempWatch.

The control group received standard care alone. At our hospital, standard care of healthy newborns involves hourly temperature measurements for the first four hours of life. If the neonate was mildly hypothermic (36.0–36.4 °C), conservative warming measures, including adding clothing layers and SSC, were implemented. If these measures did not resolve the hypothermia or their temperature was less than 36.0 °C, additional measures were implemented, such as placement under a radiant warmer or incubator, or admission to the SCN. These measures were maintained until the neonate’s temperature stabilised within the normal range (36.5–37.5 °C). On discharge, parents were recommended to regularly monitor their baby’s temperature with an axillary thermometer, although this was not enforced. Parents were also visited by a midwife within a few days after discharge (routine hospital procedure), who provided further advice on the thermal care of their baby.

### 2.6. Study Outcomes

The primary outcome was hypothermia requiring escalation of care to invasive warming measures during the neonate’s initial hospital stay after birth. Invasive warming measures included a heated mattress, radiant warmer, incubator, and/or SCN admission. The primary outcome included any hypothermic episodes requiring escalation of care from the time of randomisation until the neonate was discharged from hospital. If neonates required admission to the SCN or NICU for any reason, the primary outcome was recorded until their admission.

Pre-specified secondary outcomes included neonatal morbidities (hypothermia not requiring escalation of care, hypoglycaemia, or jaundice requiring phototherapy), interventions required (antibiotics, intravenous fluids, nasogastric feeding, or SCN/NICU admission), length of hospital stay, and time required to gain birth weight. Other secondary outcomes included the neonate’s weight, type of feeds (exclusive breastfeeding), and hours of SSC per day at discharge from hospital and 28 days of life. At 28 days, data were also collected regarding presentations and readmissions to hospital after discharge, and all-cause mortality. Additional data were collected from those allocated to the intervention group regarding their experience with the BEMPU TempWatch. This included level of compliance and free text feedback regarding their experience and any issues they encountered while using the device. All of the data were collected from online patient records and a follow-up phone call made to parents at 28 days of life.

### 2.7. Sample Size and Analysis

A previous study at our institution reported that the overall rate of hypothermia was 21–57% in neonates after 32 weeks’ gestation [14]. We therefore estimated the baseline rate of neonatal hypothermia requiring escalation of care to be 30% in our study population. To reduce this rate to 15% with 80% power and an alpha error of 0.05, a minimum sample size of 268 neonates (134 in each study group) was needed.

We planned to conduct an interim analysis after 60 babies were randomised to confirm if baseline rates of hypothermia requiring escalation of care were consistent with the expected 30%. Any safety or other concerns related to the TempWatch were also investigated at this time. It was planned that if hypothermia rates were more than 10% different from what was expected, we would adjust the sample size accordingly. If the new sample size needed was over 300 babies, the study would be deemed not feasible. Trial stopping rules were determined a priori, with the two criteria for stopping the trial being a very low event rate or any significant adverse events associated with TempWatch use.

Statistical analysis was performed using Stata (v16, StataCorp LLC, College Station, TX, USA) and GraphPad Prism (v9.2.0 for Windows, GraphPad Software, San Diego, CA, USA). Analysis of the primary and secondary outcomes was conducted by one of the trial investigators who was not involved with patient recruitment and randomisation, and data collection, and who was blinded to infants’ group allocation (KT). Data were analysed according to an intention-to-treat principle with statistical significance set at *p* < 0.05. Categorical outcomes were analysed using the Chi-square or Fisher’s exact tests and were presented as numbers with percentage proportions and a relative risk (RR). Continuous outcomes were first checked for normality using the Shapiro–Wilk test, and then analysed using the unpaired t-test or Mann-Whitney U test (as appropriate). Data were presented as means with standard deviations and difference between the means if normally distributed, or medians with interquartile ranges (IQR) and difference between the medians if not normally distributed.

## 3. Results

At the interim analysis, after reviewing the primary outcome data for the first 60 neonates, a decision was made to stop the trial as planned a priori. This was due to the low rate of hypothermia requiring escalation of care (3/60 (5.0%)) across the enrolled infant population, with the new sample size of 1810 exceeding the set feasibility threshold.

In total, during the recruitment period, 256 neonates who were LBW and/or SGA with gestational ages of 35 weeks or more were born at Monash Medical Centre. Of these, 69 neonates were admitted to the SCN or NICU soon after birth and were not returned to the postnatal wards within 48 h after delivery. Of the remaining 187 neonates who met the inclusion criteria, we approached 92 families for participation in the study with a consent rate of 81.5%. Reasons for non-inclusion are detailed in the Consolidated Standards of the Reporting Trials diagram (Figure 2). A total of 75 neonates were included in this study, with 36 randomised to the TempWatch group and 39 to the control group. Two families allocated to the control group withdrew their consent after their baby’s discharge from hospital. One family in the TempWatch group was unable to be contacted for the follow-up phone call. Data for the primary outcome and some secondary outcomes were still available for these three neonates and were included in the final analyses. Baseline characteristics of enrolled infants are detailed in Table 1 and show that the two groups were evenly matched for all maternal and neonatal characteristics.

Results for the primary and secondary outcomes are summarised in Table 2. Three neonates experienced the primary outcome, hypothermia requiring escalation of care between randomisation, and discharge from hospital. Two (5.56%) were from the TempWatch group and one (2.56%) was from the control group (RR 2.17, 95% CI 0.21 to 22.89, *p* = 0.61 using Fisher’s exact test). All three neonates required SCN admission for their hypothermia. The control group neonate also required placement in an incubator to resolve the hypothermia, while one of the TempWatch group neonates had multiple episodes of hypothermia requiring placement under a radiant warmer and in an incubator.

The TempWatch group had a significantly higher rate of exclusive breastfeeding at discharge compared with the control group (22/36 (61.1%) vs. 13/39 (33.3%), RR 1.83, 95% CI 1.10 to 3.07, *p* = 0.02 using the Chi-square test). The median (IQR) hours of SSC per day at discharge were 2.25 (1.00–5.00) and 2.00 (1.00–3.75) in the TempWatch and control group, respectively (*p* = 0.62). At 28 days of life, the rates of exclusive breastfeeding (48.6% vs. 54.1%, *p* = 0.64) and SSC (1.5 (0.50–2.50) vs. 2.00 (0.50–4.00), *p* = 0.25) were similar between the TempWatch and control group. The groups were similar for all other secondary outcomes at first hospital discharge and 28 days of life (Table 2).

Parental compliance in using the TempWatch varied. The breakdown of TempWatch use after discharge was as follows: never (2.9%), rarely (40.0%), sometimes (31.4%), mostly (8.6%), and always (17.1%). Adverse events reported by parents included some baby discomfort (25.7%) and a temporary rash (8.6%).

## 4. Discussion

In this randomised controlled trial, we investigated the benefit of the BEMPU TempWatch in reducing the rate of hypothermia requiring escalation of care in LBW/SGA late preterm and term infants. Unfortunately, there was a lack of feasibility in conducting the study for this outcome in our setting due to low event rates. However, we observed a secondary benefit of improved exclusive breastfeeding rates in the group using the TempWatch.

To the best of our knowledge, the rate of hypothermia requiring escalation of care in any setting has not been investigated using the BEMPU TempWatch prior to this study. We recruited participants up to 48 h after delivery. However, a high proportion of neonates had hypothermia prior to randomisation (58% in TempWatch group and 64% in control group). This is similar to the results of other studies, which showed that rates of neonatal hypothermia were the highest during the first few hours of life [15], because the sudden transition from the warm intrauterine environment to cooler extrauterine conditions leads to rapid cooling of the neonate soon after birth [4].

It is also important to consider the high-income setting in which the study was conducted when assessing the benefit of the TempWatch in reducing hypothermia rates, especially as the device was designed for use within resource-limited settings [9]. At our hospital, midwives regularly check babies’ temperatures and parents are routinely educated on thermal care for newborns. Within low-income settings, healthcare workers often underestimate the prevalence of neonatal hypothermia [16], and parents are inadequately educated about neonatal thermal care by healthcare workers [13]. Hence, the results of our study may not reflect the potential benefits of the TempWatch observed in LMICs.

Interestingly, we observed a significantly higher rate of exclusive breastfeeding at discharge in the TempWatch group (61%) compared with the control group (33%). This is similar to findings reported in an unpublished study (Parmar, unpublished 2020). Potential explanations for this benefit relate to the device’s role in promoting improved parent–infant bonding. The increased rates of breastfeeding with TempWatch use may be through the device’s benefit in promoting SSC [11,17], which has been shown to promote exclusive breastfeeding [18]. In our study, the rate of SSC while in hospital was slightly higher (though not statistically significant) in the TempWatch group (2.3 h/day) compared with the control group (2.0 h/day). Another potential explanation may be that the TempWatch increases paternal involvement in caring for the baby, which has been shown to play a large role in a mother’s decision to breastfeed [19]. While our study did not assess paternal involvement as a secondary outcome, anecdotal feedback regarding TempWatch use in Papua New Guinea reported that the novelty of the device increased the involvement of fathers in caring for their baby [20]. Equal rates of breastfeeding at 28 days (49% in TempWatch group vs. 54% in control group) may be due to low compliance rates once families were discharged home. Given our relatively small sample size, the increased rate of exclusive breastfeeding at discharge in the TempWatch group may have been a chance finding. Exclusive breastfeeding has been shown to reduce the incidence of neonatal hypothermia and help protect the neonate against sepsis, respiratory tract infections, and meningitis [21]. Therefore, the potential role of the TempWatch in promoting breastfeeding is clinically relevant and an area that should be further explored in larger studies.

The strengths of our study included that it was the first to investigate the benefits of TempWatch use in a high-income setting, and our study focused on neonates in the postnatal wards and followed them up after discharge home, given that the TempWatch is marketed for use by parents and carers in a community setting [9]. The planned interim analysis allowed us to discontinue the study and avoid wasting additional resources. Limitations included the lack of blinding of participants and researchers due to a lack of placebo watch. However, given the objective nature of our primary outcome (the rate of hypothermia requiring escalation of care), we believe that the lack of blinding introduced minimal risk of bias in our primary outcome. The relatively high rate of hypothermia before randomisation suggests that future studies using the TempWatch should aim to recruit either antenatally or immediately after birth to ensure infants are enrolled as soon as possible after birth.

## 5. Conclusions

We demonstrated a lack of feasibility in conducting a study using the BEMPU TempWatch to reduce hypothermia-related escalation of care for LBW and SGA neonates in our setting. We observed a potential benefit of the TempWatch in promoting exclusive breastfeeding rates at the time of hospital discharge, which needs to be explored further.

## Figures and Tables

**Figure 1 children-08-01068-f001:**
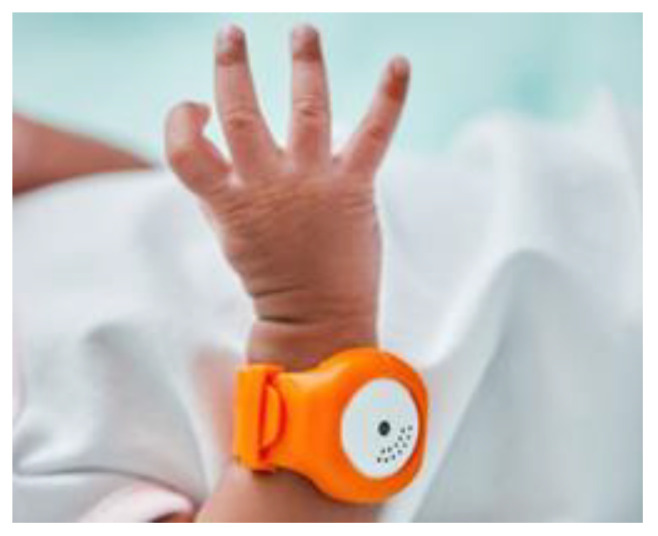
BEMPU TempWatch worn around a neonate’s wrist. Reproduced with permission from BEMPU [8].

**Figure 2 children-08-01068-f002:**
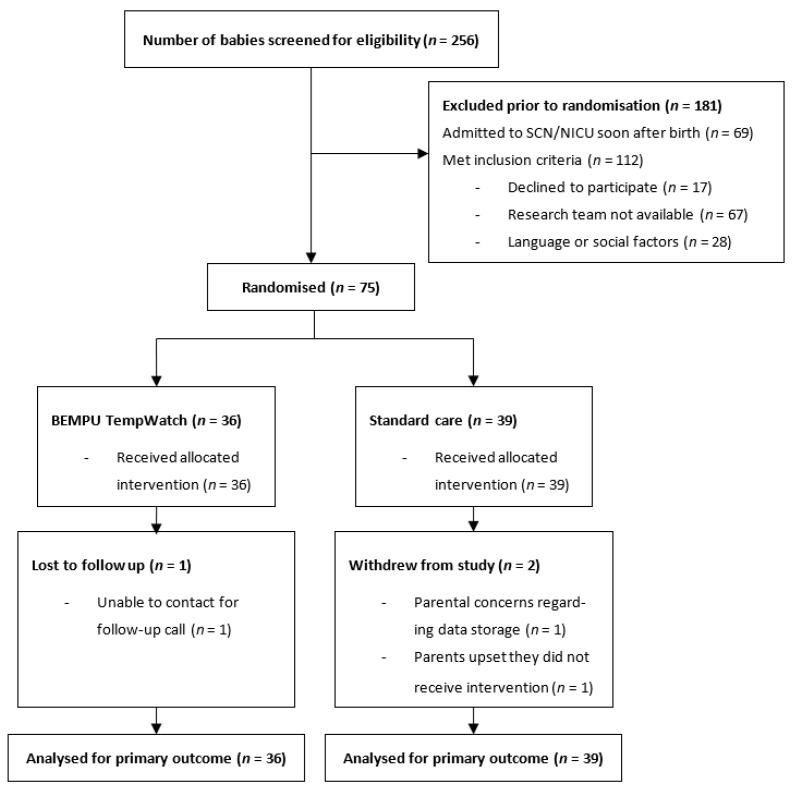
CONSORT diagram. CONSORT, Consolidated Standards of Reporting Trials.

**Table 1 children-08-01068-t001:** Demographic characteristics of mothers and neonates.

Characteristic	TempWatch Group (*n* = 36)	Control Group (*n* = 39)
**Maternal**		
Age (years)	31.4 (5.3)	31.3 (5.1)
Primiparous	17 (47.2)	16 (41.0)
Caesarean section	15 (41.7)	17 (43.6)
Premature rupture of membranes	3 (8.3)	9 (23.1)
Gestational diabetes	10 (27.8)	13 (33.3)
Pre-eclampsia	4 (11.1)	1 (2.6)
Smoking during pregnancy	2 (5.6)	1 (2.6)
Antepartum haemorrhage	4 (11.1)	4 (10.3)
**Neonatal**		
Gestational age (weeks)	38.1 (1.5)	38.0 (1.7)
Premature (<37 weeks)	8 (22.2)	10 (25.6)
Birth weight (g)	2534.2 (257.9)	2524.4 (295.3)
Low birth weight	15 (41.7)	18 (46.2)
Small for gestational age	30 (83.3)	31 (79.5)
Length (cm)	46.8 (1.7)	47.0 (2.0) ^Ŧ^
Head circumference (cm)	32.5 (1.6)	32.7 (1.6)
Male sex	21 (58.3)	24 (61.5)
Multiple birth	5 (13.9)	6 (15.4)
Apgar score, 5 min	9 (9–9)	9 (9–9)
Resuscitation required at birth	8 (22.2)	15 (38.5)
Tactile stimulation and/or suction only	5 (13.9)	10 (25.6)
IPPV and/or CPAP	3 (8.3)	5 (12.8)
Postnatal age at randomisation (h)	25.3 (9.0)	25.3 (9.7)
Hypoglycaemia prior to randomisation	12 (33.3)	21 (53.8)
Hypothermia prior to randomisation	21 (58.3)	25 (64.1)
Not requiring escalation of care	14 (38.9)	19 (48.7)
Requiring escalation of care ^†^	7 (19.4)	6 (15.4)

Data presented as *n* (%), mean (SD), or median (IQR); SD, standard deviation; IQR, interquartile range. ^†^ Escalation of care defined as requiring any of the following: heated mattress, radiant warmer, incubator and/or SCN admission. ^Ŧ^ Field contains missing data. Headings indicated in bold.

**Table 2 children-08-01068-t002:** Primary and secondary outcomes.

	TempWatch Group (*n* = 36)	Control Group (*n* = 39)	Crude Ratio (95% CI)	*p*-Value
**Primary outcome**				
Hypothermia requiring escalation of care	2 (5.6)	1 (2.6)	2.17 (0.21–22.89) *	0.61
**Secondary outcomes**				
Hypothermia not requiring escalation of care	4 (11.1)	5 (12.8)	0.87 (0.25–2.98) *	1.00
Hypoglycaemia	1 (2.8)	1 (2.6)	1.08 (0.07–16.69) *	1.00
Jaundice requiring phototherapy	5 (13.9)	11 (28.2)	0.49 (0.19–1.28) *	0.16
Interventions				
Antibiotics	0 (0.0)	2 (5.1)	0.00 (−) *	0.49
Intravenous fluids	0 (0.0)	2 (5.1)	0.00 (−) *	0.49
Nasogastric feeding	3 (8.3)	1 (2.6)	3.25 (0.35–29.85) *	0.35
SCN/NICU admission	3 (8.3)	4 (10.3)	0.81 (0.20–3.38) *	1.00
Length of hospital stay (days)	3 (2–4)	4 (3–5)	−1.00 (−1.00–0.00) ^†^	0.21
Exclusive breastfeeding at discharge	22 (61.1)	13 (33.3)	1.83 (1.10–3.07) *	0.02
Average SSC at discharge (h/day)	2.25 (1.00–5.00) ^Ŧ^	2.00 (1.00–3.75) ^Ŧ^	0.25 (−0.50–1.00) ^†^	0.62
Weight at discharge (g)	2371.77 (236.42) ^Ŧ^	2360.97 (260.84) ^Ŧ^	10.80 (−122.45–144.06) ^♦^	0.87
Time to gain birth weight (days)	7.38 (1.76) ^Ŧ^	7.42 (2.09) ^Ŧ^	−0.04 (−0.95–0.87) ^♦^	0.93
Rate of weight gain from discharge to 28 days (g/kg/day)	15.42 (12.10–22.05) ^Ŧ^	15.81 (10.68–18.06) ^Ŧ^	−0.39 (−2.57–4.79) ^†^	0.72
Exclusive breastfeeding at 28 days	17 (48.6) ^Ŧ^	20 (54.1) ^Ŧ^	0.90 (0.57–1.41) *	0.64
Average SSC at 28 days (h/day)	1.50 (0.50–2.50) ^Ŧ^	2.00 (0.50–4.00) ^Ŧ^	−0.50 (−1.50–0.50) ^†^	0.25
Presentations after discharge	10 (28.6) ^Ŧ^	8 (21.6) ^Ŧ^	1.32 (0.59–2.96) *	0.50
Readmissions after discharge	2 (5.7) ^Ŧ^	3 (8.1) ^Ŧ^	0.70 (0.13–3.97) *	1.00

Data presented as *n* (%), mean (SD), or median (IQR); SD, standard deviation; IQR, interquartile range 95% CI, 95% confidence interval; SSC, skin-to-skin contact; SCN, special care nursery; NICU, neonatal intensive care unit. * Relative risk, ^†^ Difference in median, ^♦^ Difference in mean, ^Ŧ^ Field contains missing data. Headings indicated in bold.

## Data Availability

Data available upon reasonable request.

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
