# Peer review of "Decreasing Hypothermia-Related Escalation of Care in Newborn Infants Using the BEMPU TempWatch: A Randomised Controlled Trial"

_children, 2021, doi:10.3390/children8111068_

Round 1

Reviewer 1 Report

1.Authors have taken the need for escalation of care because of hypothermia to be 30% based on their previous study. Did they include relatively more mature infants in the study or the environmental temperature was different between the two studies?

2.Authors included infants more than 35 weeks gestation and less than 1/3 were preterms. These infants may not require extra temperature support unless the environmental temperature is low.

3.Significantly higher number of infants were ob breast feeding in BEMPU group. What is the reason?

4.Usually percentages are expressed in single decimals. Sentences are not usually started with numbers and if unavoidable then they should be spelled.

5.Large number of infants are not included in the study and reason for exclusion for all of them not shown in the flow chart.

Author Response

Reviewer 1

  1. Authors have taken the need for escalation of care because of hypothermia to be 30% based on their previous study. Did they include relatively more mature infants in the study or the environmental temperature was different between the two studies?

We have updated the paper to include more information regarding the study population of the Young et al. study (page 5, section 2.7 Sample size and analysis, lines 204-205). The Young et al. study reported that the rate of hypothermia was 21-57% in neonates over 32 weeks gestation, whilst our study population included those over 35 weeks gestation. Furthermore, our study reported on the rates of hypothermia both requiring and not requiring escalation of care, whilst the Young et al. study did not differentiate between the two. Given that the two studies were conducted at the same institution, the environmental temperatures should not have varied greatly between the studies.

  1. Authors included infants more than 35 weeks gestation and less than 1/3 were preterms. These infants may not require extra temperature support unless the environmental temperature is low.

Given that the BEMPU TempWatch is targeted for use in community settings for neonates who are being cared for by parents/carers, our study population of neonates on the postnatal wards was chosen to reflect this. This is mentioned on page 9, section 4 Discussion, lines 418-420. At our hospital, neonates who are less than 35 weeks gestation are admitted to the special care nursery soon after birth, and therefore our study population only included those who were over 35 weeks gestation.

  1. Significantly higher number of infants were of breast feeding in BEMPU group. What is the reason?

Potential explanations for the higher rates of exclusive breastfeeding in the BEMPU group have been mentioned in the discussion (page 9, section 4 Discussion, lines 398-413). One potential explanation is through the device’s role in promoting skin-to-skin contact, which has been shown to promote exclusive breastfeeding. Whilst there was no significant difference in the rates of skin-to-skin contact between the groups in our study, the rate was slightly higher in the BEMPU group at discharge from hospital. The benefit may also be through the device’s role in increasing paternal involvement in caring for the baby, which has also been shown to promote breastfeeding. Our study did not assess paternal involvement as an outcome, however anecdotal feedback from Papua New Guinea reported that paternal involvement was increased with TempWatch use.

  1. Usually percentages are expressed in single decimals. Sentences are not usually started with numbers and if unavoidable then they should be spelled.

We have updated the paper so that percentages are expressed in single decimals, and to ensure that sentences do not begin with numbers.

  1. Large number of infants are not included in the study and reason for exclusion for all of them not shown in the flow chart.

The paper has been updated to include reasons for exclusion of infants from the study (pages 2-3, section 2.2 Patients, lines 121-137). For those who were admitted to the special care nursery soon after birth, reasons for admission included prematurity of less than 35 weeks gestation, birth weight less than 2000g, and significant morbidities that were unable to be managed on the postnatal wards. The reason for nursery admission for each individual neonate was not recorded, and hence unable to be included in the flow chart. Infants were also not recruited if the research team was unavailable during the time the infant was eligible for recruitment. Furthermore, the research team did not approach families if parents did not have a sufficient level of English, or if significant social factors may have impacted parental care of their baby.

Reviewer 2 Report

This is well designed RCT to determine the effect of the device. Unfortunately, the result was not positive, but I feel this is due to the well furnished setting in developed country.  The authors acknowledge the limitation of the study well.

Since they find the statistical significance in secondary outcome, it might be OK to recommend this point more detaildly.

Furthermore, for the sake of explanation of better mother to child bonding, the usability of this device should be clarified. Please attach the picture of this device as supplement, and explain why this could result in increased EBF. 

Author Response

Reviewer 2

This is well designed RCT to determine the effect of the device. Unfortunately, the result was not positive, but I feel this is due to the well furnished setting in developed country.  The authors acknowledge the limitation of the study well.

  1. Since they find the statistical significance in secondary outcome, it might be OK to recommend this point more detaildly.

The paper has been updated to elaborate on the device’s potential role in promoting exclusive breastfeeding. The TempWatch’s potential benefit in promoting parent-infant bonding has been included (page 9, section 4 Discussion, lines 400-402). The benefits of breastfeeding and its clinical relevance as a benefit of TempWatch use has also been added (page 9, section 4 Discussion, lines 413-416).

  1. Furthermore, for the sake of explanation of better mother to child bonding, the usability of this device should be clarified. Please attach the picture of this device as supplement, and explain why this could result in increased EBF. 

A picture of the BEMPU TempWatch has been attached (page 2, Figure 1, lines 98-100), along with the flowchart that was provided to families who were randomised to the intervention (TempWatch) group (page 4, Figure 2, lines 167-169). As mentioned and explained in the flowchart, parents were encouraged to provide skin-to-skin contact whenever the device indicated that their baby was hypothermic. Thus, TempWatch use may have promoted skin-to-skin contact, which has been shown to encourage exclusive breastfeeding. This potential explanation, along with other potential explanations for the increased rate of exclusive breastfeeding in the TempWatch group, has been discussed in the paper (page 9, section 4 Discussion, lines 398-413).

Round 2

Reviewer 1 Report

1.Abstract- Sentences are not started with numbers and percentage to be given in single decimals

2.Reference for sample size calculation is not appropriate since the earlier study was conducted among infants with lesser weight and gestation.

3.What is the relation between high income and temperature control?. Authors may conclude that among the infants studied ( infants > 35 weeks gestation and >1500 g birth weight) hypothermia is uncommon in their setting. 

4.Breast feeding at discharge in the Bempu group is almost double ( 61% Vs 33%). Explanation given by authors are not satisfactory. 

5.Fig.2 can be deleted

Author Response

Reviewer 1

  1. Abstract- Sentences are not started with numbers and percentage to be given in single decimals

We have checked the abstract again and have ensured that no sentences begin with numbers and all percentages in the results section are provided in single decimals.

  1. Reference for sample size calculation is not appropriate since the earlier study was conducted among infants with lesser weight and gestation.

We acknowledge that the differing populations between the Young et al. study and our study limited the accuracy of our initial sample size calculations. However, to the best of our knowledge, the Young et al. study is the only study that provides local data regarding neonatal hypothermia rates. Given that hypothermia mitigation methods and protocols vary between different settings, and hence hypothermia rates differ between hospitals, the Young et al. study was the most appropriate for our sample size calculations despite the limitations.

  1. What is the relation between high income and temperature control?. Authors may conclude that among the infants studied ( infants > 35 weeks gestation and >1500 g birth weight) hypothermia is uncommon in their setting. 

The prevalence and management of neonatal hypothermia varies between high- and low-income countries due to the differing resources, education, and local protocols in place. Hence, it was important to identify whether the benefits of the TempWatch shown in low- and middle-income countries were also applicable to a high-income setting. We have included a section in the manuscript to clarify this (page 2, section 1 Introduction, lines 108-111). In our study, the rate of hypothermia was 46/75 (61.3%) before and 12/75 (16.0%) after randomisation. This shows that, despite our study population consisting of relatively more mature and larger neonates, and the study being conducted within a well-resourced tertiary hospital, hypothermia rates remained high during the neonatal period.

  1. Breast feeding at discharge in the Bempu group is almost double (61% Vs 33%). Explanation given by authors are not satisfactory. 

We have further updated our discussion surrounding the increased rate of exclusive breastfeeding at discharge in the BEMPU group. We have included a sentence recognising that the difference between the groups may have been a chance finding due to the relatively small sample size of the study (page 8, section 4 Discussion, lines 406-408) and have recommended that this potential benefit be further explored in larger studies (page 8, section 4 Discussion, line 411).

  1. 2 can be deleted

Thank you for the suggestion, we have now deleted Figure 2 (the flowchart of care).

Round 3

Reviewer 1 Report

Authors mention that the percentage of hypothermia is more than 30% in their setting. Could they detect these hypothermia cases using BEMPU watch?. There was no difference in escalation in the two groups as mentioned by authors. Hence most of them should have been cases of transient hypothermia after birth. Authors may conclude that there is no requirement of this watch in the setting studied rather than considering the income of the country. The risk of hypothermia among the cases recruited is likely to be low unless the environmental temperature is low.

Author Response

Authors mention that the percentage of hypothermia is more than 30% in their setting. Could they detect these hypothermia cases using BEMPU watch?. There was no difference in escalation in the two groups as mentioned by authors. Hence most of them should have been cases of transient hypothermia after birth. Authors may conclude that there is no requirement of this watch in the setting studied rather than considering the income of the country. The risk of hypothermia among the cases recruited is likely to be low unless the environmental temperature is low.

Thank you for the suggestion. Included it in the conclusions